# READ BEFORE IMPUTING:
# INJECTING PUBMED-INFORMED SEMANTIC PRIORS FOR MULTI-TISSUE GENE EXPRESSION IMPUTATION

## ABSTRACT

The integration of gene expression across tissues and cell types is essential for uncovering the systemic biological mechanisms that underlie disease and homeostasis. Yet in practice, gene expression data are rarely available for all tissues, posing a major barrier to understanding cross-tissue regulation and disease etiology. Existing methods attempt to overcome this issue by imputing tissue-specific gene expression from the observed expression of other tissues. However, these methods rely solely on observed data while overlooking biological priors—fundamental knowledge sources that can critically enhance biologically meaningful predictions. To address this limitation, we propose ReadImpute, a novel framework for multi-tissue gene expression imputation that injects semantic priors derived from biomedical literature into the imputation process. ReadImpute leverages retrieval-augmented generation (RAG) with a local large language model (LLM) to distill PubMed articles into semantic embeddings of genes and tissues, which serve as external priors guiding a neural network for multi-tissue gene expression imputation. Extensive experimental results demonstrate that ReadImpute significantly improves imputation performance and generalizes well to unseen tissue profiles. ReadImpute bridges the gap between biomedical literature and data-driven learning, offering a biologically grounded solution for gene expression imputation.

## 1 INTRODUCTION

Most complex diseases manifest through dysfunction across multiple tissues and organs (Consortium, 2020). For example, hypertension involves metabolic changes in various organs, including the heart, blood vessels, brain, and kidneys (Dai et al., 2018). These dysfunctions are often driven and reflected in transcriptional changes occurring across different tissues (Cookson et al., 2009; Nica et al., 2010). In other words, transcriptional variation plays a central role in mediating the causal links between genetic variation and complex traits. Therefore, analyses of tissue-specific gene expression profiles can offer valuable insights into disease etiology and support patient stratification (Grant et al., 2002). However, since collecting gene expression data from many tissues requires invasive procedures (e.g., biopsies), expression is typically measured only in accessible tissues like blood, resulting in unobserved data for many tissues.

To address this issue, several methods have been developed to impute tissue-specific gene expression based on measurements from accessible tissues. TEEBoT (Basu et al., 2021) attempts to predict tissue-specific gene expression in other tissues using only whole blood transcriptome data. More recently, HYFA (Viñas et al., 2023) performs tissue-specific imputation by leveraging observed expression data from multiple tissues. These methods train models to infer unobserved tissue-specific expression from observed data by minimizing reconstruction error.

Despite their potential, a fundamental limitation of existing approaches is their exclusive reliance on observed gene expression data. These methods treat the task as a purely data-driven machine learning problem, without incorporating biological knowledge about individual genes or tissues. Specifically, gene and tissue embeddings are randomly initialized in an undifferentiated manner. However, there are biological relationships (e.g., associating and dissociating) among genes (Whelan et al., 2020; Hall et al., 2021). Furthermore, tissues also exhibit biologically meaningful relationships, such as functional similarity or anatomical proximity (Pierson et al., 2015).

To address this limitation, we propose ReadImpute, a novel approach for multi-tissue gene expression imputation that integrates literature-informed knowledge into the imputation process. ReadImpute collects domain knowledge from PubMed (Sayers et al., 2021), which contains numerous biomedical articles, and generates literature-informed embeddings for both genes and tissues. To enable this, ReadImpute utilizes retrieval-augmented generation (RAG) (Lewis et al., 2020) with a local large language model (LLM) (Touvron et al., 2023) to extract textual summaries that capture the biological context of each gene or tissue. These summaries are then encoded into embedding vectors using a pretrained sentence encoder (Reimers & Gurevych, 2019). The literature-informed gene embeddings are incorporated into the encoding process of gene expression, while the tissue embeddings are utilized within a hypergraph neural network that models the interactions among individuals, tissues, and genes. This seamlessly integrated literature-informed knowledge enables the model to make more biologically grounded predictions, significantly improving imputation performance. Extensive experimental results demonstrate the superiority of ReadImpute over existing methods in both multi-tissue gene expression imputation and cell-type signature prediction.

In summary, our key contributions are as follows: 1) To the best of our knowledge, ReadImpute is the first approach to incorporate literature-informed domain knowledge into multi-tissue gene expression imputation. 2) By injecting semantic priors derived from biomedical literature, ReadImpute substantially enhances performance in both mutli-tissue gene expression imputation and cell-type signature prediction. 3) By bridging the gap between gene expression imputation and biomedical literature, we present a reusable framework for extracting biological priors via RAG and sentence-level encoding, which can be applied to integrate literature-derived knowledge into task-specific neural architectures across diverse biological problems.

## 2 RELATED WORK

### 2.1 IMPUTATION OF GENE EXPRESSION DATA

Since missing data is a prevalent issue across various domains, its imputation has long been a central problem in machine learning (Allison, 2009). A wide range of methods have been developed, from simple techniques such as zero imputation (Schafer & Graham, 2002) and k-nearest neighbors (kNN) imputation (Troyanskaya et al., 2001) to more sophisticated statistical approaches (Van Buuren & Groothuis-Oudshoorn, 2011; Li & Li, 2018). With the advancement of deep learning, deep learning-based approaches have gained popularity due to their effectiveness in imputing missing data (Yoon et al., 2018; You et al., 2020). In computational biology, a variety of methods have been proposed to impute diverse types of biological data (Eraslan et al., 2019; Wang et al., 2021; Choi et al., 2023). For transcriptomic data imputation, TEEBoT (Basu et al., 2021) predicts tissue-specific gene expression from whole blood using linear models. More recently, HYFA (Viñas et al., 2023) frames the task as multi-tissue gene expression imputation using a hypergraph-based model, enabling flexible inference from arbitrary subsets of observed tissues. Despite their practicality, both TEEBoT and HYFA rely solely on observed gene expression data, without leveraging external biological knowledge, which leads to suboptimal performance under sparse observations.

### 2.2 LLMS FOR COMPUTATIONAL BIOLOGY

The rise of LLMs (Devlin et al., 2019; Brown et al., 2020) has spurred growing interest in their application to biomedical problems. Models such as BioBERT (Lee et al., 2020) and PubMedBERT (Gu et al., 2021), and recent GPT (Brown et al., 2020)-style architectures pretrained on biomedical corpora (Luo et al., 2022) have shown promising results across a wide range of biological tasks. However, fine-tuning LLMs with external data remains costly and resource-intensive (Singh et al., 2024). RAG offers a practical alternative by retrieving relevant information from external knowledge sources and integrating it into the generation process (Lewis et al., 2020). Recently, GENERAG (Lin et al., 2024) attempt to answer gene-related questions by incorporating external genetic knowledge into LLMs through a RAG-based approach. Similarly, scRAG (Yu et al., 2025) uses hybrid retrieval to guide LLMs for single-cell annotation, while GenePT (Chen & Zou, 2025) directly constructs cell embeddings by averaging GPT-generated gene embeddings. Compared to these approaches, which either employ LLMs as reasoning engines or as direct embedding generators, our ReadImpute introduces a distinct paradigm: it converts biomedical literature into semantic priors and integrates them into a task-specific neural network, thereby guiding biologically grounded representation learning.

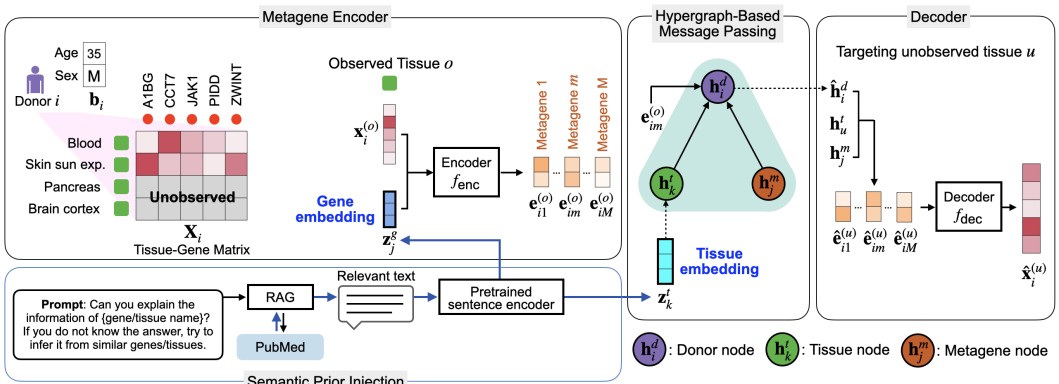

Figure 1: An overview of ReadImpute.

# 3 PROPOSED METHOD

## 3.1 PROBLEM SETUP

We aim to impute missing tissue-specific gene expression profiles using observed multi-tissue transcriptomic data collected from individuals (donors). Let $\mathcal{I} = \{1, \ldots, N\}$ denote the set of individuals. Each individual $i \in \mathcal{I}$ is associated with a demographic feature vector $\mathbf{b}_i \in \mathbb{R}^C$ and a tissue-gene expression matrix $\mathbf{X}_i \in \mathbb{R}^{T \times G}$, where $C$ is the number of covariates, $T$ is the number of tissues, and $G$ is the number of genes. In practice, however, only a subset of tissues is profiled for each individual, resulting in partially observed rows in $\mathbf{X}_i$. Let $\mathcal{I}_{\text{train}}$, $\mathcal{I}_{\text{val}}$, and $\mathcal{I}_{\text{test}}$ be a partition of $\mathcal{I}$, representing the training, validation, and test sets, respectively. For each individual $i \in \mathcal{I}$, let $\mathbf{x}_i^{(t)} \in \mathbb{R}^G$ denote the gene expression vector for tissue $t \in \{1, \ldots, T\}$, and let $\mathcal{O}(i) \subseteq \{1, \ldots, T\}$ be the index set of tissues for which expression is observed. We define the partially observed matrix $\widetilde{\mathbf{X}}_i \in (\mathbb{R} \cup \{*\})^{T \times G}$ such that the $t$-th row $\widetilde{\mathbf{x}}_i^{(t)}$ equals $\mathbf{x}_i^{(t)}$ if $t \in \mathcal{O}(i)$, and consists entirely of missing entries (denoted by $*$) otherwise. Formally, for each training individual $i \in \mathcal{I}_{\text{train}}$, the model is trained to approximate the conditional distribution $p(\mathbf{X}_i \mid \widetilde{\mathbf{X}}_i, u_i)$ in order to infer the missing tissue-specific gene expression values based on the observed subset. The ultimate goal of multi-tissue gene expression imputation is to accurately recover unobserved tissue-specific gene expression profiles for individuals $i' \in \mathcal{I}_{\text{test}}$, given their partially observed tissue profiles $\widetilde{\mathbf{X}}_{i'}$ and covariates $u_{i'}$.

## 3.2 OVERVIEW OF READIMPUTE

For multi-tissue gene expression imputation, we present ReadImpute, a novel framework that integrates literature-informed knowledge into the imputation process. Figure 1 illustrates an overview of ReadImpute, which comprises four stages: semantic prior injection, metagene encoding, hypergraph-based message passing, and decoding. The semantic prior injection stage generates prior-injected semantic embeddings for both genes and tissues by leveraging a large language model (LLM) with retrieval-augmented generation (RAG) on biomedical literature (see Sec. 3.3). The metagene encoder stage produces metagene embeddings using both the gene expression profiles from observed tissues and the semantic embeddings of genes (see Sec. 3.4). Next, the hypergraph-based message passing stage incorporates tissue-level semantic priors into the message passing process to update the representations of donors, tissues, and metagenes (see Sec. 3.5). Finally, the decoder predicts the gene expression of unobserved tissues based on these updated representations, yielding the final output of ReadImpute (see Sec.3.6).

## 3.3 SEMANTIC PRIOR INJECTION

To inject domain knowledge from biomedical literature into the imputation process, we develop a unified module that integrates document retrieval, LLM-based RAG, and sentence-level embedding. This module produces literature-informed semantic embeddings for both genes and tissues, which serve as external priors that provide inductive bias toward biologically grounded predictions.

We access biomedical literature by querying the PubMed database (Sayers et al., 2021) via the NCBI Entrez API (Sayers et al., 2021), and retrieve relevant documents for each gene or tissue. For genes, we construct search queries using both the official symbol and the full descriptive name of each gene, the latter obtained from the NCBI Entrez Gene database. For tissues, we use their common names directly as query terms. We then extract the abstract section from each retrieved document. Each retrieved abstract is encoded into a fixed-length $d$-dimensional vector using a pretrained sentence-level encoder (Reimers & Gurevych, 2019), resulting in a set of abstract embeddings.

To efficiently handle and query a large number of abstract embeddings, we employ FAISS (Facebook AI Similarity Search) (Douze et al., 2024), a library designed for fast similarity search. We construct a FAISS-based vector store from these embeddings and use a local LLM (Touvron et al., 2023) to perform retrieval-augmented generation (RAG), where the LLM receives a gene or tissue name as input, retrieves the most relevant abstracts from the vector store, and synthesizes a concise textual summary that describes the entity's biological function or role. These summaries are subsequently encoded into $l_z$-dimensional vectors using the same sentence-level encoder, and serve as the final literature-informed semantic embeddings. The semantic embedding for gene $j$ is denoted by $\mathbf{z}_j^g \in \mathbb{R}^{l_z}$. Similarly, the semantic embedding for tissue $k$ is denoted by $\mathbf{z}_k^t \in \mathbb{R}^{l_z}$. $\mathbf{z}_j^g$ and $\mathbf{z}_k^t$, containing knowledge from PubMed, are incorporated into the metagene encoder stage and the hypergraph-based message passing stage, respectively.

## 3.4 METAGENE ENCODER

The metagene encoder aims to extract a compact and informative representation of high-dimensional gene expression by transforming raw expression values into a low-dimensional set of metagene embeddings. In ReadImpute, the metagene encoder generates metagene embeddings by leveraging not only observed tissue-specific gene expression but also semantic priors associated with each gene.

Formally, let $\mathbf{x}_i^{(o)} \in \mathbb{R}^G$ be the gene expression vector of observed tissue $o$ from individual $i$. The metagene encoder computes two representations. First, the expression vector $\mathbf{x}_i^{(o)}$ is directly mapped to metagene space via a single-layer linear encoder $f_{\text{expr}}$:

$$\mathbf{e}_{i,\text{expr}}^{(o)} = f_{\text{expr}}(\mathbf{x}_i^{(o)}) \in \mathbb{R}^{M \cdot l_m}, \tag{1}$$

where $M$ is the number of metagenes and $l_m$ is the metagene dimension.

Second, we generate semantic prior-injected representations using $\{\mathbf{z}_j^g\}_{j=1}^G$. We first define the gene embedding matrix $\mathbf{Z}^g \in \mathbb{R}^{G \times l_z}$ by stacking the column vectors $\{\mathbf{z}_j^g\}_{j=1}^G$ horizontally and taking the transpose:

$$\mathbf{Z}^g = \begin{bmatrix} \mathbf{z}_1^g & \mathbf{z}_2^g & \cdots & \mathbf{z}_G^g \end{bmatrix}^\top \in \mathbb{R}^{G \times l_z}. \tag{2}$$

We then project the expression vector $\mathbf{x}_i^{(o)}$ into the semantic embedding space using $\mathbf{Z}^g$:

$$\mathbf{w}_i^{(o)} = \mathbf{x}_i^{(o)} \mathbf{Z}^g \in \mathbb{R}^{l_z}, \tag{3}$$

and the resulting $\mathbf{w}_i^{(o)}$ is passed through a separate single-layer linear encoder $f_{\text{sem}}$:

$$\mathbf{e}_{i,\text{sem}}^{(o)} = f_{\text{sem}}(\mathbf{w}_i^{(o)}) \in \mathbb{R}^{M \cdot l_m}. \tag{4}$$

The final metagene embedding is obtained by averaging the two outputs and reshaping:

$$\mathbf{E}_i^{(o)} = \text{reshape}\left(\frac{1}{2}(\mathbf{e}_{i,\text{expr}}^{(o)} + \mathbf{e}_{i,\text{sem}}^{(o)})\right) \in \mathbb{R}^{M \times l_m}. \tag{5}$$

These metagene embeddings $\mathbf{E}_i^{(o)}$, injected with gene-level semantic priors, are used as hyperedge features in the subsequent hypergraph-based message passing stage.

## 3.5 HYPERGRAPH-BASED MESSAGE PASSING

Inspired by HYFA (Viñas et al., 2023), we develop a hypergraph-based imputation model to capture the complex interactions among individuals, tissues, and metagenes. ReadImpute constructs a three-uniform hypergraph where each hyperedge connects a donor $i$, an observed tissue $o \in \mathcal{O}(i)$, and a

metagene $m \in \{1, \ldots, M\}$. Each hyperedge is associated with a metagene feature vector $\mathbf{e}_{im}^{(o)} \in \mathbb{R}^{l_m}$, denoting the $m$-th row of $\mathbf{E}_i^{(o)}$ and capturing both expression-level and gene-level semantic information.

When constructing the hypergraph that consists of three types of nodes (*i.e.*, donor, tissue and metagene), node features are initialized based on their type. Let $\mathbf{h}_i^d(0) \in \mathbb{R}^{n_d}$, $\mathbf{h}_k^t(0) \in \mathbb{R}^{n_t}$, and $\mathbf{h}_j^m(0) \in \mathbb{R}^{n_m}$ denote the initial node features for donor, tissue, and metagene nodes. Donor nodes are initialized using the demographic feature vector $\mathbf{b}_i$ (e.g., age and sex), *i.e.*, $\mathbf{h}_i^d(0) = \mathbf{b}_i$ and $n_d = C$. For each metagene node, $\mathbf{h}_j^m(0)$ is randomly initialized with trainable embeddings that are updated through message passing. For tissue nodes, we first prepare trainable embeddings. For tissue $k \in \{1, \ldots, T\}$, we first randomly initialize a trainable embedding $\tilde{\mathbf{h}}_k^t(0)$ for its node. We then incorporate semantic priors derived from biomedical literature: each semantic prior-injected tissue embedding $\mathbf{z}_o^t \in \mathbb{R}^{l_z}$ is linearly transformed via a trainable weight matrix $W_{\text{tissue}} \in \mathbb{R}^{n_t \times l_z}$ and added to $\tilde{\mathbf{h}}_k^t(0)$ to generate initial features $\mathbf{h}_k^t(0)$ for the tissue node:

$$\mathbf{h}_k^t(0) = \tilde{\mathbf{h}}_k^t(0) + W_{\text{tissue}}\mathbf{z}_k^t. \tag{6}$$

This process ensures that the initial representation of each tissue node integrates both a learnable component and semantic information obtained from biomedical literature.

Since donor embeddings are initially based solely on demographic features, we perform message passing over the hypergraph to update donor representations using information from each donor's tissues and genes. Message passing is carried out through $L$ stacked hypergraph neural network layers. During this process, only donor embeddings are updated, while tissue and metagene embeddings remain fixed to preserve independence across donors. Formally, at each layer, we update donor embeddings as follows:

$$\hat{\mathbf{h}}_i^d = \phi_h(\mathbf{h}_i^d, \mathbf{m}_i) \in \mathbb{R}^{\hat{n}_d}, \tag{7}$$

$$\mathbf{m}_i = \sum_{m=1}^{M} \sum_{o \in \mathcal{O}(i)} \phi_a(\mathbf{h}_m^m, \mathbf{h}_o^t, \mathbf{m}_{im}^{(o)}) \in \mathbb{R}^{n_{mes}}, \tag{8}$$

$$\mathbf{m}_{im}^{(o)} = \phi_e(\mathbf{h}_i^d, \mathbf{h}_m^m, \mathbf{h}_o^t, \mathbf{e}_{im}^{(o)}) \in \mathbb{R}^{n_{mes}}, \tag{9}$$

where $\phi_e$ is an edge function that combines node embeddings and hyperedge features to produce messages, $\phi_a$ is an aggregation function that determines how messages are pooled for each donor node, and $\phi_h$ is a node update function. The edge function $\phi_e$ and node update function $\phi_h$ are implemented as multi-layer perceptrons (MLPs).

The aggregation function $\phi_a$ is implemented using a learnable attention mechanism that assigns different weights to messages based on the relevance of each metagene–tissue pair to the donor. Specifically, the attention-based aggregation is defined as:

$$\phi_a(\mathbf{h}_m^m, \mathbf{h}_o^t, \mathbf{m}_{im}^{(o)}) = \alpha_{mo} \cdot \mathbf{m}_{im}^{(o)}, \tag{10}$$

where the attention coefficient $\alpha_{mo}$ is computed as:

$$\alpha_{mo} = \frac{\exp(e_{mo})}{\sum_v \exp(e_{vo})}, \tag{11}$$

$$e_{mo} = \mathbf{a}^\top \text{LeakyReLU}(W[\mathbf{h}_m^m \| \mathbf{h}_o^t]). \tag{12}$$

Here, $W \in \mathbb{R}^{n_{att} \times (n_m + n_t)}$ and $\mathbf{a} \in \mathbb{R}^{n_{att}}$ are learnable parameters, and $\|$ denotes vector concatenation.

We denote the updated donor embedding at the first layer as $\hat{\mathbf{h}}_i^d(1)$, and at the $a$-th layer as $\hat{\mathbf{h}}_i^d(a)$. The final output of hypergraph-based message passing stage is $\hat{\mathbf{h}}_i^d(L)$, injected with tissue-level semantic priors via $\mathbf{z}_k^t$.

## 3.6 DECODER AND OPTIMIZATION

To impute the gene expression vector $\hat{\mathbf{x}}_i^{(u)}$ of an unobserved tissue $u$ for donor $i$, we first predict metagene attributes:

$$\hat{\mathbf{e}}_{im}^{(u)} = \text{MLP}([\hat{\mathbf{h}}_i^d(L) \| \mathbf{h}_t \| \mathbf{h}_m]) \in \mathbb{R}^{l_m}, \tag{13}$$

Table 1: Impuation performance comparison on the GTEx-v8 dataset, measured by the Pearson correlation.

| Tissue | Mean | kNN | TEEBoT | HYFA (Blood) | HYFA (All) | ReadImpute (Blood) | ReadImpute (All) |
|---|---|---|---|---|---|---|---|
| Adipose Visceral Omentum | -0.039 | 0.365 | 0.395 | 0.486 | 0.508 | 0.496 | **0.517** |
| Artery Aorta | 0.010 | 0.323 | 0.377 | 0.459 | 0.485 | 0.470 | **0.494** |
| Artery Tibial | 0.110 | 0.326 | 0.358 | 0.420 | 0.431 | 0.423 | **0.438** |
| Breast Mammary Tissue | 0.036 | 0.334 | 0.376 | 0.375 | 0.378 | 0.389 | **0.409** |
| Cells Cultured | 0.248 | 0.234 | 0.139 | 0.284 | 0.333 | 0.323 | **0.360** |
| Colon Sigmoid | -0.047 | 0.314 | 0.246 | 0.493 | 0.511 | 0.481 | **0.525** |
| Colon Transverse | 0.027 | 0.297 | 0.238 | 0.322 | 0.344 | 0.328 | **0.379** |
| Esophagus Gastro | 0.027 | 0.392 | 0.403 | 0.529 | 0.548 | 0.517 | **0.565** |
| Esophagus Mucosa | 0.031 | 0.329 | 0.320 | 0.434 | 0.460 | 0.435 | **0.466** |
| Esophagus Muscularis | 0.025 | 0.393 | 0.411 | 0.544 | 0.566 | 0.561 | **0.582** |
| Heart Atrial | -0.073 | 0.380 | 0.366 | 0.475 | 0.495 | 0.503 | **0.521** |
| Lung | 0.049 | 0.355 | 0.379 | 0.472 | 0.479 | 0.484 | **0.493** |
| Muscle Skeletal | 0.023 | 0.436 | 0.482 | 0.518 | 0.525 | 0.522 | **0.527** |
| Nerve Tibial | -0.049 | 0.356 | 0.395 | 0.459 | 0.455 | 0.466 | **0.467** |
| Pituitary | 0.038 | 0.147 | 0.140 | 0.273 | 0.295 | 0.253 | **0.317** |
| Stomach | -0.051 | 0.060 | 0.153 | 0.171 | 0.190 | 0.189 | **0.227** |
| Testis | -0.034 | 0.110 | 0.161 | 0.259 | 0.227 | 0.259 | **0.282** |
| Thyroid | 0.009 | 0.332 | 0.309 | 0.428 | 0.463 | 0.435 | **0.464** |
| **Average** | 0.019 | 0.305 | 0.314 | 0.411 | 0.428 | 0.423 | **0.447** |

for each metagene $m \in 1, \ldots, M$. We then construct the predicted metagene representation:

$$\hat{\mathbf{E}}_i^{(t)} = \begin{bmatrix} \hat{\mathbf{e}}_{i1}^{(t)} & \hat{\mathbf{e}}_{i2}^{(t)} & \cdots & \hat{\mathbf{e}}_{iM}^{(t)} \end{bmatrix}^{\top} \in \mathbb{R}^{M \times l_m}, \tag{14}$$

which is flattened and decoded into the gene space:

$$\hat{\mathbf{x}}^{(t)}i = f_{\text{dec}}(\text{flatten}(\hat{\mathbf{E}}^{(t)}i)) \in \mathbb{R}^G, \tag{15}$$

where $f_{\text{dec}}$ is a feedforward decoder shared across all tissues.

To train the model, we adopt a pseudo-masking strategy to simulate the imputation of unobserved tissues. For each training individual $i$, we randomly select a subset of their observed tissues $\mathcal{O}'(i) \subset \mathcal{O}(i)$ and treat the remaining observed tissues $\mathcal{U}'(i) = \mathcal{O}(i) \setminus \mathcal{O}'(i)$ as pseudo-missing. The model is trained to reconstruct the gene expression profiles $\mathbf{x}_i^{(t)}$ for each $t \in \mathcal{U}'(i)$, corresponding to the pseudo-missing tissues, using only the pseudo-observed subset $\mathcal{O}'(i)$. This procedure effectively increases the number of training examples and promotes generalization. We minimize the average mean squared error (MSE) across the pseudo-missing tissues:

$$\mathcal{L}(\widetilde{\mathbf{X}}_i, \mathbf{b}_i, \mathcal{U}'(i)) = \frac{1}{|\mathcal{U}'(i)|} \sum_{t \in \mathcal{U}'(i)} |\hat{\mathbf{x}}_i^{(t)} - \mathbf{x}_i^{(t)}|_2^2. \tag{16}$$

At test time, we apply the trained model to individuals with partially observed tissues to impute gene expression profiles for their unobserved tissues using all available information.

## 4 EXPERIMENTS

### 4.1 EXPERIMENTAL SETUP

For comparison, we consider four baseline models, including state-of-the-art methods developed for multi-tissue gene expression imputation: (1) Mean (Graham et al., 1997) imputes missing tissue expression by averaging the gene expression of that tissue across all individuals in the training set; (2) k-NN imputation (Troyanskaya et al., 2001) predicts missing values by averaging gene expression from the $k$ most similar individuals; (3) TEEBoT (Basu et al., 2021) leverages pretrained transcriptomic embeddings and linear regression to infer gene expression in unobserved tissues; and (4) HYFA (Viñas et al., 2023) utilizes a hypergraph-based encoder-decoder architecture to model cross-tissue relationships and impute unobserved gene expression. We conduct experiments on the GTEx-v8 and GTEx-v9 datasets (Consortium, 2020). Since our study our study covers multiple types of experiments, we describe the corresponding setups in the following subsections. Additional implementation details including dataset preprocessing, hyperparameters, training procedure, and semantic prior generation, are provided in Appendix A.

### 4.2 IMPUTATION PERFORMANCE COMPARISON

**GTEx-v8** To evaluate the performance of our proposed ReadImpute, we adopt the experimental protocol established by HYFA, which utilizes the GTEx-v8 dataset for tissue-level gene expression

imputation. In this setup, a subset of tissues is masked per individual, and models are tasked with imputing the missing gene expression values. Evaluation is conducted under two conditions: (i) Accessible, where only tissues that were originally collected for a given individual are used as targets for imputation, and (ii) All, where all GTEx-defined tissues are treated as potential targets regardless of whether they were measured. As shown in Table 1, under both conditions, ReadImpute consistently achieves the best or competitive performance compared to HYFA and other baselines, including Mean, k-NN, and TEEBoT. In particular, ReadImpute exhibits strong imputation accuracy in the accessible setting, underscoring its robustness when evaluated on observed data. Overall, ReadImpute outperforms all baseline methods across both evaluation settings.

**GTEx-v9** We further assess the generalizability of ReadImpute by comparing it to HYFA on the GTEx-v9 dataset, focusing on the model's ability to recover cell-type–specific expression programs. To this end, we freeze the encoder and hypergraph layers of the pretrained models and reinitialize only the decoder to predict expression signatures for ten reference cell types (e.g., adipocyte, T cell) derived from GTEx-v9 single-nucleus RNA-seq data. We then fine-tune the decoder for 300 epochs using a learning rate of $1{\times}10^{-3}$ and evaluate the Pearson correlation coefficient (PCC) between the predicted and ground truth single-nucleus signatures. As shown in Table 2, ReadImpute surpasses HYFA in cell type–level accuracy, achieving a higher overall PCC (from 0.729 to 0.763). These results demonstrate that incorporating literature-informed metagene priors enhances the biological fidelity of imputed signals beyond bulk-level inference alone.

Table 2: Imputation performance comparison on the GTEx-v9 dataset, measured by the Pearson correlation.

| Tissue | HYFA | ReadImpute |
|---|---|---|
| Adipocyte | 0.613 | **0.700** |
| Endothelial cell (lymphatic) | 0.804 | **0.850** |
| Endothelial cell (vascular) | 0.795 | **0.826** |
| Fibroblast | 0.745 | **0.792** |
| Immune (B cell) | 0.623 | **0.638** |
| Immune (DC / macrophage) | 0.733 | **0.747** |
| Immune (NK cell) | 0.784 | **0.799** |
| Immune (T cell) | 0.794 | **0.821** |
| Immune (mast cell) | 0.633 | **0.672** |
| Pericyte / SMC | 0.770 | **0.789** |
| **Average** | 0.729 | **0.763** |

## 4.3 ABLATION STUDY

To investigate the contribution of gene- and tissue-level prior knowledge to ReadImpute's performance, we conduct an ablation study summarized in Table 3. Specifically, we evaluate three ablated variants of our model: (i) a version without gene and tissue priors, (ii) a version without gene priors, and (iii) a version without tissue priors. These variants are compared against the full ReadImpute model under both accessible and all evaluation settings.

Table 3: Ablation study.

| | Accessible | All |
|---|---|---|
| ReadImpute (w/o Gene, Tissue) | 0.411 | 0.428 |
| ReadImpute (w/o Gene) | 0.422 | 0.441 |
| ReadImpute (w/o Tissue) | 0.420 | 0.443 |
| ReadImpute | **0.423** | **0.447** |

The results show that removing either gene or tissue priors consistently reduces performance, confirming that both types of literature-informed priors contribute complementary information. Notably, ablating tissue-level priors results in a larger drop in the all setting (from 0.447 to 0.441), while removing gene-level priors yields a comparable decrease in the accessible setting (from 0.423 to 0.422). The largest degradation is observed when both priors are removed, indicating a strong synergistic effect between gene- and tissue-level representations. Overall, the full ReadImpute model achieves the highest accuracy in both settings, highlighting the benefit of incorporating structured biological priors for both genes and tissues.

## 4.4 PREDICTION OF CELL TYPE SIGNATURES

A cell type signature is a $G$-dimensional vector constructed by averaging single-cell or single-nucleus RNA-seq expression values for each gene $g$ across all cells belonging to a given cell type $c$ (e.g., adipocyte, endothelial cell). Reconstructing such signatures from bulk data provides a strong test of whether a model captures fine-grained, cell-type–specific transcriptional programs—beyond explaining global expression variance at the tissue level.

To assess this, we construct reference signatures for three representative cell types using GTEx-v9 single-nucleus RNA-seq data: adipocyte, endothelial cell (lymphatic), and fibroblast. Expression values are $\log_2(\text{TPM} + 1)$–transformed and Z-score normalized on a per-gene basis prior to aver-

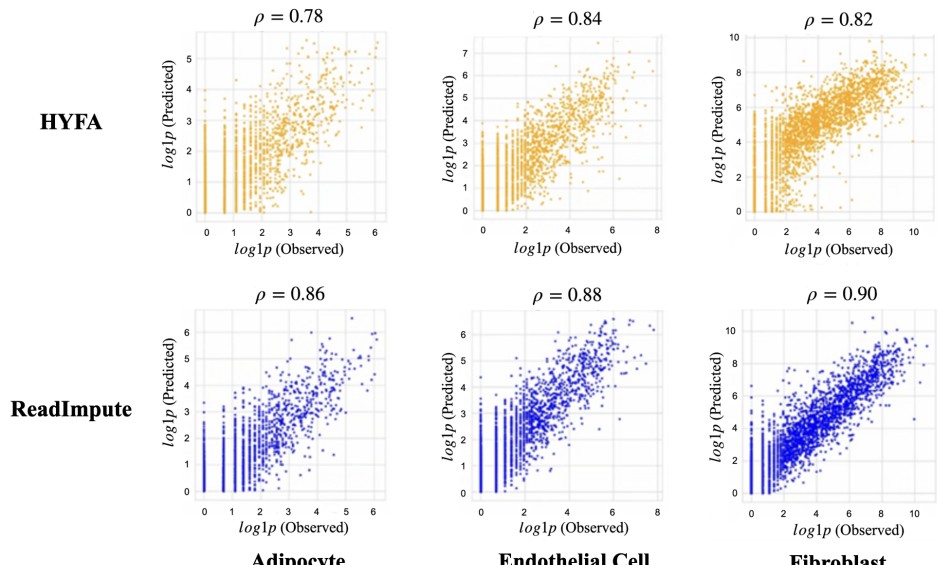

Figure 2: Cell type signatures of HYFA (left) and ReadImpute (right).

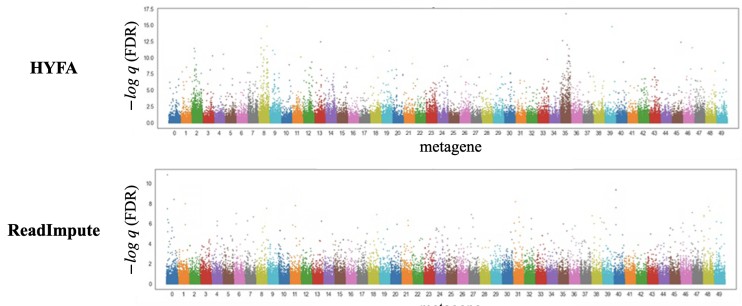

Figure 3: Manhattan plot of the Gene Set Enrichment Analysis (GSEA).

aging. For each donor-tissue pair $(p, t)$, the model predicts a gene expression vector $\hat{x}_{p,t}$, which is then compared against the corresponding cell type signature via Pearson correlation.

Figure 2 presents scatter plots for HYFA (left) and ReadImpute (right) across three representative tissues. The ReadImpute predictions align more closely with the ground-truth signatures, as evidenced by tighter clustering along the 45° diagonal. This highlights the inductive benefit of literature-informed priors, as well as the role of the orthogonality constraint in preserving diverse and disentangled transcriptional features across cell types.

### 4.5 PATHWAY ENRICHMENT ANALYSIS OF METAGENE-FACTORS

Metagenes are low-dimensional representations that capture co-expression patterns of genes. To ensure that these representations are biologically meaningful, we assess whether the learned metagene factors align with known functional pathways. To this end, we conduct pathway enrichment analysis using Gene Set Enrichment Analysis (GSEA) over all metagene-factor pairs ($50 \times 98 = 4,900$ combinations), ranking genes by their factor loadings. The KEGG pathway database serves as the reference gene set collection, and a pathway is considered significantly enriched if it passes the False Discovery Rate (FDR) threshold ($q$-value $< 0.05$) after Benjamini–Hochberg correction.

As illustrated in Figure 3, ReadImpute achieves broader coverage across metagenes, whereas HYFA displays a sparser pattern, with significant enrichments concentrated in some metagenes. This contrast suggests that literature-informed priors in ReadImpute facilitate more modular specialization of biological processes within the latent space. The resulting alignment with KEGG pathways improves interpretability and demonstrates ReadImpute's ability to recover structured functional programs from high-dimensional gene expression data.

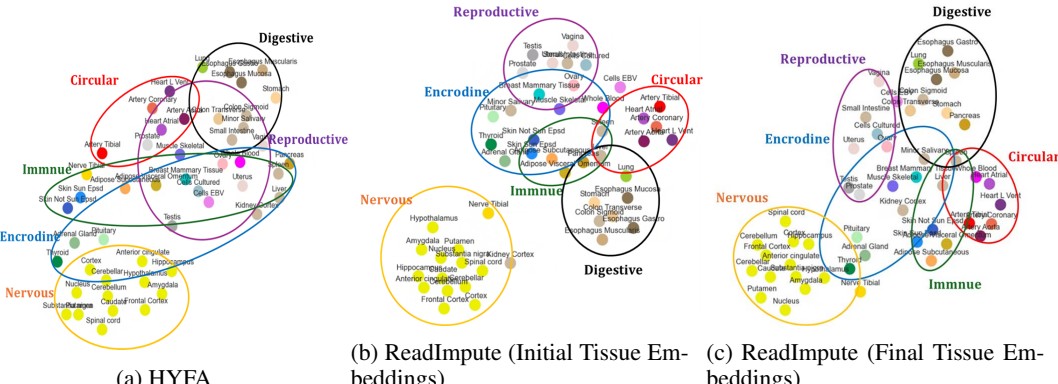

|(a) HYFA|(b) ReadImpute (Initial Tissue Embeddings)|(c) ReadImpute (Final Tissue Embeddings)|

Figure 4: Comparison of tissue embeddings from (a) HYFA, (b) ReadImpute (Initial Tissue Embedding $\mathbf{z}^t$), and (c) ReadImpute (Tissue Embedding $\mathbf{h}^t$).

## 4.6 TISSUE EMBEDDING ANALYSIS

To investigate whether ReadImpute captures biologically meaningful relationships among tissues, we visualize the tissue embeddings produced by HYFA, the initial tissue embeddings of ReadImpute, and the final tissue embeddings of ReadImpute using t-SNE in Figure 4. The initial tissue embeddings are derived from a Large-Language Model, while the final embeddings incorporate gene-tissue expression data through the ReadImpute training process. Each tissue is colored according to its system-level biological classification (e.g., nervous, cardiovascular, endocrine, immune, digestive, reproductive), allowing a systematic evaluation of functional clustering quality.

We highlight three key observations: (1) **Improved Functional Clustering:** Compared to HYFA, both the initial and final tissue embeddings of ReadImpute exhibit improved clustering of tissues based on shared biological functions. This indicates that the literature-informed priors effectively encodes semantic-level biological knowledge even before training. (2) **Accurate Placement of Multi-Functional Tissues:** ReadImpute accurately embeds multi-functional tissues such as the liver and spleen. Unlike HYFA, which often places such tissues in functionally isolated regions, ReadImpute locates them near the intersection of relevant functional clusters. This reflects their multiple roles in immune, metabolic, and endocrine functions. (3) **Biologically Plausible Localization of Neuroencrodyne Tissues:** While the literature-informed initial embeddings of neuroendocrine tissues (e.g., thyroid, pituitary) fail to reflect their biological proximity to the nervous system, the final embeddings produced by ReadImpute place them in more biologically plausible positions, adjacent to neural clusters. As shown in Figure 4(a), these neuroendocrine tissues are embedded close to nervous system tissues, suggesting that gene–tissue expression signals enable ReadImpute to recover this biologically coherent alignment.

These findings collectively validate that ReadImpute effectively integrates gene expression signals with literature–driven biological knowledge, resulting in tissue embeddings that are not only functionally coherent but also aligned with established biological hierarchies. Further experimental results, including single-source imputation (Appendix B.1) and additional gene-set enrichment analysis (Appendix B.2), are provided in Appendix B.

## 5 CONCLUSION

We introduce ReadImpute, a novel framework that enhances multi-tissue gene expression imputation by integrating literature-informed domain knowledge. Unlike previous methods that rely solely on observed gene expression data, ReadImpute generates semantic embeddings derived from PubMed using RAG, a local LLM, and a pretrained sentence-level encoder. These PubMed-informed semantic embeddings are then seamlessly integrated into the imputation process. We demonstrate that incorporating external biological knowledge from literature substantially improves performance in both gene expression imputation and cell-type signature prediction tasks. Beyond its performance gains, ReadImpute provides a generalizable and modular framework for infusing literature-derived knowledge into biological tasks. We believe this opens new avenues for bridging the gap between data-driven learning and biomedical literature. Extending the ReadImpute framework to other computational biology tasks with abundant yet underutilized domain knowledge is left for future work.

## Reproducibility Statement

We have made efforts to ensure the reproducibility of our work. Detailed descriptions of the datasets, preprocessing pipelines, model architectures, hyperparameters, and training procedures are provided in Appendix A. We also include the prompt design and pipeline for generating literature-informed semantic priors in the same appendix. The GTEx datasets used in this study are publicly available through the GTEx portal. Our implementation is based on standard libraries such as PyTorch Geometric and SentenceTransformers, and all random seeds are fixed to ensure consistent results across runs. The complete source code will be made publicly available upon publication.

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

# A EXPERIMENTAL DETAILS

## A.1 DATASET DETAILS

We conduct experiments on the GTEx dataset, a publicly available resource comprising gene expression data collected form diverse human tissues. We download the dataset from the GTEx portal[1]. Specifically, we use the GTEx-v8 bulk RNA-seq dataset for multi-tissue gene expression imputation and the GTEx-v9 single-nucleus RNA-seq (snRNA-seq) dataset for cell type signature prediction. For both datasets, we follow the preprocessing protocols described in (Viñas et al., 2023).

### A.1.1 GTEx-v8

Following the GTEx-v8 eQTL discovery pipeline[2], we apply the following preprocessing steps:

- We discard under-represented tissues (n = 5), including bladder, cervix (ectocervix and endocervix), fallopian tube, and kidney (medulla).
- We retain a shared set of protein-coding genes across all tissues.
- We remove donors with only one sampled tissue (n = 4).
- We apply an expression filter, keeping genes with $\geq 0.1$ TPM in $\geq 20\%$ of samples and $\geq 6$ raw counts in $\geq 20\%$ of samples.
- We normalize raw counts using the trimmed mean of M-values (TMM) method.
- We apply per-gene inverse normal transformation across samples.

The final dataset contains 15,197 samples from 834 donors across 49 tissues and 12,557 genes. Donors are randomly split into 500 for training, 167 for validation, and 167 for testing. On average, each donor has 18.22 tissue samples.

### A.1.2 GTEx-v9

We utilize paired single-nucleus RNA-seq (snRNA-seq) data from 16 GTEx donors collected across eight tissues (skeletal muscle, breast, oesophagus mucosa and muscularis, heart, lung, prostate, and skin). We retain overlapping genes between the bulk and single-cell datasets, and select the top 3,000 highly variable genes. Cell types appearing in fewer than 10 tissue–individual pairs are discarded. Read counts are aggregated by donor, tissue, and broad cell type, resulting in 226 unique cell-type signatures.

## A.2 IMPLEMENTATION DETAILS

We conduct experiments on a single NVIDIA GeForce RTX 3090 Ti GPU and an Intel Core i9-12900k CPU.

**Model Configuration.** ReadImpute builds upon the default architectural settings of HYFA (Viñas et al., 2023), with modifications to incorporate semantic prior injection. We leverages a GAT-style hypergraph neural network with two graph layers and multi-head attention. The key architectural hyperparameters of ReadImpute are as follows:

- Number of metagenes: $M = 50$
- Metagene embedding dimension: $l_m = 98$
- Literature-informed semantic embedding dimension: $l_z = 384$
- Donor (patient) feature dimension: 71
- Tissue feature dimension: 120
- Gene projection dimension: 48
- Edge feature dimension: 28

---

[1] https://www.gtexportal.org/home/
[2] https://github.com/broadinstitute/gtex-pipeline/tree/master/qtl

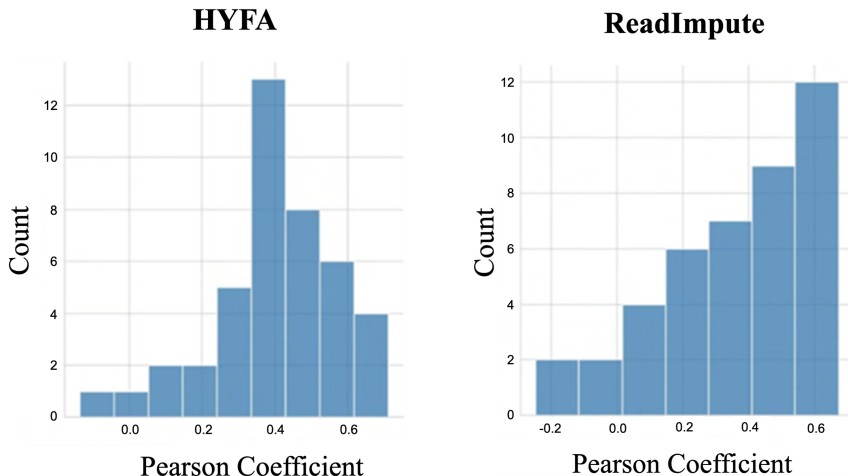

Figure 5: Pearson correlation coefficient distributions between predicted Esophagus Mucosa expression and the actual expression when only whole-blood profiles are provided.

- Number of attention heads: 28
- Batch size: 63
- Dropout rate: 0.1739

**Training Details.** We train the model for a maximum of 500 epochs with early stopping (patience = 30). We use the Adam optimizer with a learning rate of $4.56 \times 10^{-4}$ and no weight decay. The decoder and prediction MLP contain 2 hidden layers, while the message-passing network uses 1 hidden layer per block. Training is conducted with a pseudo-masking strategy for tissue-level imputation, and the loss is computed as the mean squared error (MSE) over pseudo-missing tissues.

**Semantic Prior Generation.** We obtain gene- and tissue-level semantic priors via a pipeline consisting of:

1. PubMed document retrieval using NCBI Entrez API
2. Text extraction via BeautifulSoup
3. RAG-based summarization using the locally executed `llama3.2` (Touvron et al., 2023) via Ollama[3].
4. Sentence embedding generation using the all-MiniLM-L6-v2 model (Reimers & Gurevych, 2019) from SentenceTransformers, a library built on top of HuggingFace Transformers (Wolf et al., 2020).

The prompt used for both genes and tissues is:

```
What is the [entity] in the human body?  If you do not know
the answer, try to infer it from similar [tissues/proteins].
Format your response as 1., 2., 3.  numbered items.  Be
concise and fact-focused.
```

We retrieve up to 5 documents per entity and use a FAISS-based retriever (Douze et al., 2024) to query the embedded abstracts. If no document is found, we directly query the LLM with the prompt above. The resulting 384-dimensional semantic embeddings are z-score normalized and subsequently projected into model-specific feature spaces using learnable linear layers; For instance, tissue embeddings are mapped from 384 to 120 dimensions.

---

[3]Ollama is a lightweight framework for running large language models locally. See `https://ollama.com`.

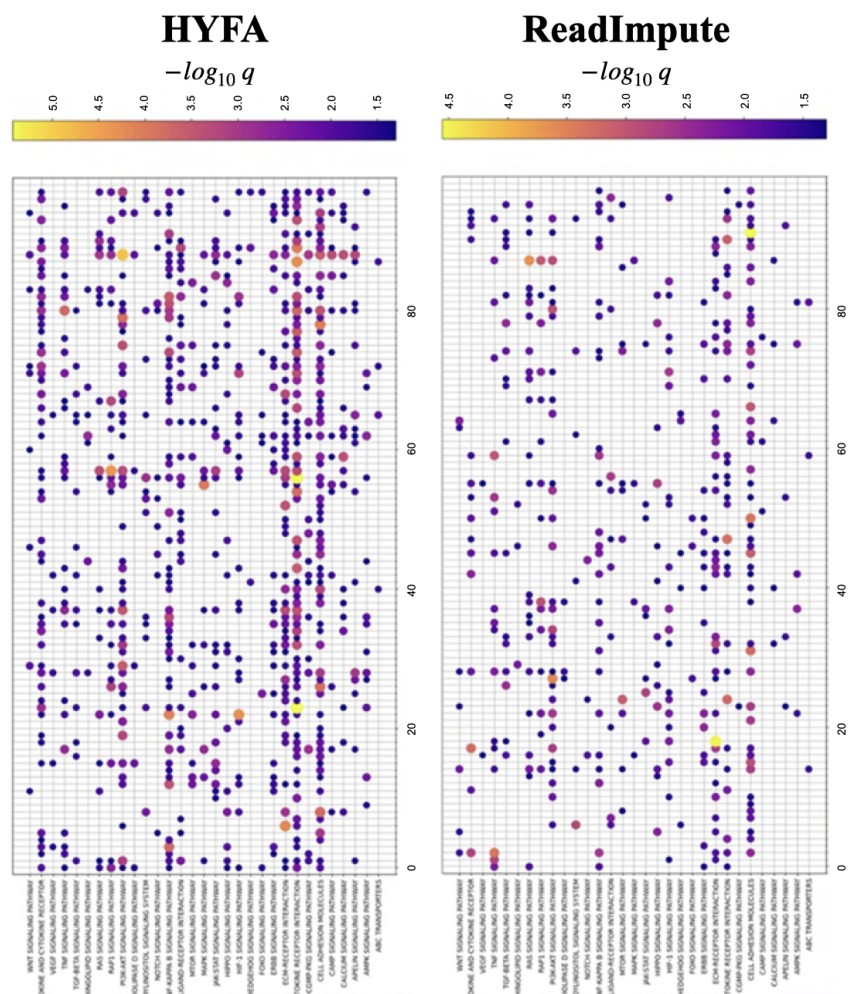

Figure 6: Gene-Set Enrichment Analysis on KEGG. ReadImpute exhibits a more localized enrichment patterns compared to HYFA.

# B    ADDITIONAL EXPERIMENTS

## B.1    SINGLE-SOURCE IMPUTATION (FROM WHOLE BLOOD TO ESOPHAGUS MUCOSA)

To test the practical scenario in which only an easily collected tissue is available, we investigate whether ReadImpute can infer the Esophagus Mucosa from whole blood alone. We filter the GTEx-v8 validation partition for donors that possessed expression profiles for all four source tissues (Whole Blood, two skin sites, subcutaneous adipose) and the target tissue Esophagus Mucosa. Only 57 donors satisfied this criterion. For those donors, we retain Whole Blood only (source) and mask the remaining tissues. Therefore, HYFA and ReadImpute do not receive information from the Esophagus Mucosa. We then compute the Pearson correlation between the predicted Esopohagus mucosa expression and the actual expression. As shown in Figure 5, ReadImpute reached a mean Pearson coefficient $\phi = 0.42$. Also, 93% of donors exceeded $\rho > 0.20$ and none were negative. In contrast, HYFA achieves $\rho = 0.18$.

These results show that a single, routinely sampled tissue can drive biologically meaningful imputation of a distant organ, more than doubling cross-tissue concordance relative to the previous state-of-the-art. Such robustness is crucial for downstream eQTL mapping and TWAS in cohorts where multi-tissue panels are incomplete.

## B.2 ADDITIONAL GENE-SET ENRICHMENT ANALYSIS

To evaluate the biological interpretability of the learned metagene representations, we conduct gene set enrichment analysis (GSEA) on KEGG signaling pathways. Specifically, for each metagene, we rank genes based on their loading values and perform GSEA using the KEGG pathway gene sets. The resulting False Discovery Rate (FDR)-adjusted p-values (-$\log_{10}$ scale) are visualized as a dot matrix where each column corresponds to a metagene and each row to a KEGG pathway. Dot color and size indicate the enrichment significance, with brighter and larger dots representing stronger associations.

Figure 6 compares the pathway enrichment profiles of metagenes learned by ReadImpute and HYFA. ReadImpute exhibits highly localized enrichment patterns: individual pathways are selectively associated with a small number of metagenes, forming sparse but distinctive activation signatures. In contrast, HYFA's metagenes tend to exhibit a bit more diffused enrichment patterns across multiple pathways, resulting in a denser but less specific activation profile. This qualitative difference suggests that ReadImpute metagenes are more semantically disentangled and less redundant.

The pathway-level sparsity and selectivity observed in ReadImpute directly enhance the interpretability of its latent space. Since each metagene aligns with a limited subset of biologically coherent pathways, downstream users can more easily attribute functional meaning to metagene activations. This disentangled structure contrasts with HYFA, where overlapping enrichment makes interpretation ambiguous. Overall, these findings demonstrate that ReadImpute produces biologically structured representations that effectively support human interpretation and pathway-level reasoning.

