# OpenReview forum: "Read Before Imputing: Injecting PubMed-Informed Semantic Priors for Multi-Tissue Gene Expression Imputation"
_ICLR.cc/2026/Conference — ICLR 2026 Conference Withdrawn Submission_

### Official Review · Reviewer_FXiE · 2025-10-28

**Soundness:** 3
**Presentation:** 2
**Contribution:** 2
**Rating:** 4
**Confidence:** 3

**Summary:**

Integrating multi-omics data across diverse cell types, tissues, and donors is inherently challenging, particularly due to missing measurements such as incomplete gene expression profiles. This paper introduces ReadImpute, an imputation framework that leverages textual embeddings of biological entities (e.g., genes and tissues) derived from large language models with retrieval-augmented generation (RAG). By incorporating these biologically informed embeddings, the method aims to enhance generalization to previously unseen tissue profiles. The approach is evaluated on the well-established GTEx-v8 and GTEx-v9 datasets, demonstrating strong performance in gene expression imputation. Beyond numeric accuracy, ReadImpute shows utility in predicting cell-type signatures and in generating biologically meaningful embeddings, as evidenced by pathway enrichment analyses and tissue-level clustering.

**Strengths:**

Leveraging prior knowledge from large language models (LLMs) in downstream, task-specific machine learning models is an important and timely research direction, given the broad and powerful conceptual representations learned by LLMs. This paper explores the novel idea of incorporating biological priors into an imputation framework, a contribution that stands out as original within this context.

The proposed method is evaluated comprehensively, assessing not only the accuracy of imputation but also the biological interpretability and meaningfulness of the learned representations.

**Weaknesses:**

The paper presents a complex imputation framework comprising three main modules; however, its relationship to existing methods—particularly HYFA, which is included in the benchmarking—is not clearly articulated.

Several important baselines appear to be missing, especially well-established imputation approaches from the machine learning community, such as missForest, MIWAE (for MAR data), and not-MIWAE (for NMAR data). Including these would provide a more rigorous comparative evaluation.

In addition, some experimental details are insufficiently described, making it difficult to fully assess the methodology. Please refer to my specific questions below for clarification.

**Questions:**

1. What is the specific contribution of this paper in terms of neural network architecture? How does it advance or differ from existing designs?

2. In the ablation study, how is the model configured without prior embeddings? Were these embeddings replaced with non-informative representations such as one-hot encodings or random vectors, or was the entire module removed? I recommend using non-informative embeddings to isolate the effect of prior embeddings and ensure that observed differences are not simply due to the presence of additional modules.

3. The distinction between the two evaluation conditions is unclear. Could the authors clarify the differences? Where does the ground truth for imputation originate? Is it feasible to evaluate the proposed model under both in-distribution and out-of-distribution imputation scenarios?

4. In the pathway enrichment analysis, HYFA’s results appear more significant than those of ReadImpute. Can the authors provide an interpretation or possible explanation for this observation?

---

### Official Review · Reviewer_65po · 2025-10-31

**Soundness:** 2
**Presentation:** 3
**Contribution:** 3
**Rating:** 4
**Confidence:** 4

**Summary:**

This paper addresses the challenge of imputing missing gene expression profiles across cell types, tissues, and donors to facilitate data integration. The authors introduce ReadImpute, an imputation framework that leverages gene and tissue embeddings derived from large language models (LLMs) to incorporate known biological relationships between genes and tissues. This design enables the model to generalize effectively to unseen tissue profiles. The proposed approach is rigorously evaluated on GTEx-v8 and GTEx-v9 datasets, as well as in predicting cell type–specific signatures. Furthermore, the biological relevance of the learned embeddings is assessed through enrichment analyses and visualization, providing evidence of their interpretability and utility in downstream applications.

**Strengths:**

•	Addresses a significant and broadly relevant problem with clear implications for the biological and biomedical research community.
•	Introduces a novel approach that leverages biological priors extracted from large language models to enhance imputation, a concept with potential applicability to a wide range of related problems.
•	Demonstrates thorough evaluation, examining the proposed method from multiple complementary perspectives to provide a well-rounded assessment of its performance and utility.

**Weaknesses:**

•	The model architecture integrates three complex components—metagene encoder, hypergraph-based message passing, and decoder—but the rationale for including each, as well as the specific choice of large language model, is not clearly articulated. Additional ablation studies would help clarify the contribution of each component.
•	While the evaluation is broad in scope, experimental results are reported without error bars, limiting the ability to assess statistical reliability. Furthermore, the description of experimental setups lacks sufficient detail, making replication and deeper interpretation challenging.

**Questions:**

•	Are all components of the proposed model novel? Could a simpler baseline (e.g., linear regression using concatenated donor, gene, and tissue embeddings) achieve similar performance? Could ablation studies be conducted to isolate the contribution of each component?
•	What is the main difference between ReadImpute and HYFA? Is ReadImpute without gene/tissue embeddings equivalent to HYFA?
•	Would alternative embeddings, such as gene representations from BioBERT or GenePT, lead to comparable results?
•	What is the specific motivation for using retrieval-augmented generation (RAG) in this setting?

---

### Official Review · Reviewer_dJL7 · 2025-11-01

**Soundness:** 3
**Presentation:** 4
**Contribution:** 3
**Rating:** 6
**Confidence:** 4

**Summary:**

This paper introduces ReadImpute, a framework that improves multi-tissue gene expression imputation by incorporating semantic priors from PubMed. The key innovation is that current imputation models such as like TEEBoT and HYFY, rely only on observed expression data and ignore biological knowledge about genes and tissues, in other words, genes and tissue embeddings in those models are randomly initialized. ReadImpute bridges this gap by using retrieval-augmented generation (RAG) with a local LLM to summarize biomedical literature for each gene and tissue. These summaries are embedded (use a pretrained sentence encoder) into vectors that serve as semantic priors, guiding a hypergraph neural network to better model relationships among genes, tissues, and donors.

The framework consists of:
1. Semantic prior injection: retrieves and encodes gene/tissue summaries via PubMed + LLM + sentence encoder.
2. Metagene encoder: fuses expression data with literature-derived gene embeddings.
3. Hypergraph message passing: incorporates tissue-level semantic priors into donor-tissue-gene relations.
4. Decoder: reconstructs unobserved tissue expression profiles.

Results
1. On GTEx-v8, ReadImpute outperforms baselines (kNN, TEEBoT, HYFA) across nearly all tissues.
2. Average Pearson correlation improves from 0.428 (HYFA-All) to 0.447 with ReadImpute.
3. Gains are especially strong for difficult tissues such as Colon and Esophagus, showing that literature priors help in low-data regimes.
4. ReadImpute generalizes well even when imputing unseen tissues, outperforming models without priors.

Evaluation
The fundamental idea of the paper is straightforward - using prior knowledge (RAG + sentence embedding) to augment observed gene expression data, i.e., bridging text and expression through semantic priors, and provided incremental improvement ob architecture (built on HYFA). It provides a meaningful step toward literature-informed omics modeling and opens a promising (and relatively low cost) direction for integrating LLMs with biological data. The novelty is clear and well-motivated, but the gains are moderate and the setup is complex.

**Strengths:**

1. Novel integration of literature priors: The paper effectively combines LLM-based knowledge retrieval (RAG on PubMed) with expression modeling, adapting proven architectures (e.g., HYFA-style hypergraph networks) rather than inventing new ones. The approach is not ground-breaking architecturally but is conceptually well-motivated and well-suited for the biomedical domain.

2. Low-dimensional metagene compression: The proposed metagene encoder is an intriguing design choice that enables efficient representation of high-dimensional gene expression while incorporating semantic priors from literature.

3. Biological grounding: The PubMed-derived embeddings provide interpretable, biologically meaningful priors that replace random initialization, improving biological fidelity and interpretability.

4. Flexible and extensible framework: The design can generalize to other omics data types (e.g., cell-type, proteomics, metabolomics) and potentially to other biomedical modalities such as small molecules or phenotypic data.

5. Consistent empirical gains: ReadImpute demonstrates stable, reproducible improvements over established baselines (e.g., HYFA, TEEBoT) on public GTEx datasets, highlighting practical relevance and robustness.

6. Comprehensive biomedical analyses: The authors include relevant downstream evaluations: cell-type, pathway, and tissue-embedding analyses that enhance the biological interpretability of their model and support its validity.

**Weaknesses:**

1. The reliance on PubMed retrieval quality introduces potential noise or bias from irrelevant abstracts.

2. Improvements, though consistent, are modest in magnitude (~2–4%), so the biological significance may need deeper validation.

3. The framework is complex, combining RAG (local LLM), FAISS, and hypergraph networks, hard to reproduce without released code.

**Questions:**

1. On the use of PubMed priors:
How sensitive is ReadImpute’s performance to the quality or quantity of retrieved PubMed abstracts?
Did you evaluate whether the LLM-generated summaries introduce noise or hallucinations that could bias the embeddings?
Are you using the entire PubMed or just the abstract? PMC? Open access? Will the license restriction significantly impact your results?
Why PubMed (although it is the obvious choice)? Have you considered/tried other sources?

2. Semantic embedding construction:
Why was a sentence encoder (sentence BERT) chosen over domain-specific models like PubMedBERT or BioGPT for embedding generation?
Are the gene and tissue semantic embeddings jointly fine-tuned during training, or frozen after generation?

3. Generalization and extensibility:
Can this approach extend to single-cell data (e.g. perturbation prediction in scRNA-seq) where literature priors might be more sparse?

**Details Of Ethics Concerns:**

No concern

---

### Official Review · Reviewer_ex2P · 2025-11-03

**Soundness:** 3
**Presentation:** 3
**Contribution:** 3
**Rating:** 4
**Confidence:** 4

**Summary:**

The authors of this paper introduce ReadImpute, a multi-tissue gene expression imputation model that incorporates literature-informed semantic priors for genes and tissues. ReadImpute uses RAG, with an LLM to query PubMed abstracts and generate concise textual summaries for genes and tissues. These summaries are then converted into semantic embeddings via a sentence encoder, which are injected into a hypergraph-based message passing module. Experiments demonstrate consistent improvements over the baselines, with higher Pearson correlations in imputation tasks, better cell-type signature prediction, and more biologically interpretable latent factors.

**Strengths:**

- The use of RAG for biological prior extraction is well-motivated. The proposed approach integrates biological information using semantic embeddings without requiring curated ontologies or external knowledge graphs.

- The experimental evaluation is extensive and convincing. The GTEx-v8 and v9 results cover both tissue-level and cell-type-level tasks. The authors also conducted ablation studies to confirm that both gene- and tissue-level priors contribute to the final performance.

**Weaknesses:**

- The comparison to alternative knowledge integration approaches is limited. The paper benchmarks against TEEBoT and HYFA but not against models that leverage ontology-based priors (e.g., gene co-function networks, STRING-based embeddings).

- The authros did not compare with pre-existing biomedical embeddings (e.g., BioWordVec, Gene2Vec embeddings). It would be interesting to compare ReadImpute with those simpler alternatives.

**Questions:**

- The authors compare ReadImpute primarily against data-driven imputation methods (TEEBoT and HYFA). However, several previous works have integrated structured biological priors, such as co-function networks (e.g., STRING, GeneMANIA) or ontology-based gene embeddings (e.g., GO2Vec). Could the authors clarify whether they considered or experimented with such graph- or ontology-based priors? If not, could they discuss why these were excluded and how they expect ReadImpute’s literature-informed priors to differ conceptually or empirically from these network-based ones?

- The paper argues for the novelty of using RAG-generated summaries and sentence embeddings, but many pre-trained biomedical embeddings already exist (e.g., BioWordVec, BioBERT-based Gene2Vec, or concept-level embeddings trained on UMLS). These models also encode semantic relationships between genes and tissues. Could the authors compare ReadImpute’s priors against these embeddings?

---

### Note · Authors · 2025-11-16

I have read and agree with the venue's withdrawal policy on behalf of myself and my co-authors.